# Low Thermal Stress and Instant Efficient Atomization of Narrow Viscous Microfluid Film Using a Paper Strip Located at the Edge of a Surface Acoustic Wave Atomizer

**DOI:** 10.3390/mi16060628

**Published:** 2025-05-27

**Authors:** Yulin Lei, Yusong Li, Jia Ning, Yu Gu, Chenhui Gai, Qinghe Ma, Yizhan Ding, Benzheng Wang, Hong Hu

**Affiliations:** 1School of Mechanical Engineering and Automation, Harbin Institute of Technology, Shenzhen 518055, China; yulinlei1@163.com (Y.L.); 23s153090@stu.hit.edu.cn (Y.L.); 20b353003@stu.hit.edu.cn (J.N.); 24b953041@stu.hit.edu.cn (Y.G.); 22b353002@stu.hit.edu.cn (C.G.); 22s153167@stu.hit.edu.cn (Q.M.); dingyizhan0414@163.com (Y.D.); 2Shenzhen Axxon Automation Co., Ltd., Shenzhen 518110, China

**Keywords:** surface acoustic wave (SAW), atomization, aerosol, liquid film, thermal stress

## Abstract

A traditional SAW (surface acoustic wave) atomizer directly supplies liquid to the surface of the atomized chip through a paper strip located in the path of the acoustic beam, resulting in irregular distribution of the liquid film, which generates an aerosol with an uneven particle size distribution and poor directional controllability, and a high heating phenomenon that can easily break the chip in the atomization process. This paper presents a novel atomization method: a paper strip located at the edge of the atomizer (PSLEA), which forms a micron-sized narrow liquid film at the junction of the atomization chip edge and the paper strip under the effect of acoustic wetting. By using this method, physical separation of the atomized aerosol and jetting droplets can be achieved at the initial stage of atomizer startup, and an ideal aerosol plume with no jetting of large droplets, a uniform particle size distribution, a vertical and stable atomization direction, and good convergence of the aerosol beam can be quickly formed. Furthermore, the effects of the input power, and different paper strips and liquid supply methods on the atomization performance, as well as the heating generation capacity of the liquid in the atomization zone during the atomization process were explored through a large number of experiments, which highlighted the advantages of PSLEA atomization. The experiments demonstrated that the maximum atomization rate under the PSLEA atomization mode reached 2.6 mL/min initially, and the maximum thermal stress was 45% lower compared with that in the traditional mode. Additionally, a portable handheld atomizer with stable atomization performance and a median aerosol particle size of 3.95 μm was designed based on the proposed PSLEA atomization method, showing the great potential of SAW atomizers in treating respiratory diseases.

## 1. Introduction

Atomization can be defined as the process of breaking up a bulk liquid into microscopic droplets, termed aerosols, by using external energy to disrupt the surface tension of the liquid. The diameter of these aerosols is typically on the micron scale (tens to hundreds of microns), which can be suspended in the air to form a mist [1]. In recent years, surface acoustic wave (SAW) atomization has been shown to be a promising and efficient atomization approach due to the directionally concentrated energy being confined in the vicinity of the solid surface in the frequency range of 10 MHz–100 MHz [2,3,4]. Although excited SAWs only have a small vibration amplitude on the nanometer scale, they can produce a vibration acceleration of up to 10^7^ m/s^2^ [5,6]. The extremely high acceleration at the boundary can directly transfer enough acoustic energy to the microfluidics to produce micron-sized monodisperse aerosols without any moving parts and nozzle restrictions, suggesting that they could have great potential in inhalation treatments for respiratory diseases, such as asthma and COPD, and in vaccine delivery [7,8,9].

In principle, fluid atomization can result from the interaction between acoustic waves on the surface of a piezoelectric substrate and a fluid placed on the acoustic beam propagation path. A stable continuous fluid supply and thin liquid film morphology are prerequisites for achieving stable atomization. Compared with other liquid supply methods, such as loading droplets through capillary tubes or pumping the liquid through a capillary slit combined with a syringe pump, the liquid delivery technology based on a paper strip is considered to be a simple, practical, and effective method and does not require an additional power supply system nor does it require cumbersome special treatments on the SAW chip [10,11]. It has become the most popular method of liquid transportation in SAW atomization. The liquid is drawn from the liquid storage tank to the surface of the SAW atomizer using a paper strip. The paper-based liquid transportation system only needs to connect one end of the paper strip to the liquid storage tank and the other end to the surface of the SAW atomizer so that the liquid can be continuously transported to the surface of the SAW atomizer through the capillary effect formed by the microporous channel of the paper strip, which forms a liquid meniscus and interacts with SAWs to achieve atomization [12,13]. However, there are still some limitations to the liquid supply method of placing the paper strip on the surface of the atomizer. If the paper strip is directly placed on the surface of the atomizer, part of the acoustic energy will leak into the paper strip after passing through the liquid meniscus in front of the paper strip, resulting in energy loss. To make matters worse, it may also cause the paper strip to curl, thus affecting the shape of the front-end liquid meniscus and adversely affecting the monodispersity and atomization rate of the atomized aerosol. In this regard, several studies have reported that the aerosol particle size from SAW atomization had a multi-peak distribution because the atomization process will always generate large particle droplets that are hundreds of microns in size [14,15,16]. In addition, the expansion of the liquid meniscus is unstable, resulting in the formation of an aerosol mist in a divergent state with an uncontrollable direction, and the deposition of large droplets on the electrode will damage the interdigital transducer (IDT). Moreover, the interaction between the SAWs and the liquid will produce an acoustic streaming heat effect, which manifests as a relatively high temperature in the atomization zone at the front of the paper strip, further forming a temperature gradient on the surface of the atomizer that could cause the atomization chip to crack [17,18,19,20].

In this paper, a sustainable and efficient atomization method with a liquid supply paper strip located at the edge of the atomizer (PSLEA) is proposed. Compared with the traditional atomization method with a liquid supply paper strip located on the surface of the atomizer (PSLSA), to allow for a continuous liquid supply using the paper strip, with PSLEA, the edge of the atomizer is in contact with the paper strip and a very short thin liquid film is formed at the edge of the atomizer. Physical separation of atomized aerosol and jetting large droplets can be achieved at the beginning of atomization. At the same time, it can quickly achieve a stable, directional, regular, uniform, and convergent aerosol plume under appropriate power conditions with the atomization direction perpendicular to the atomizer surface. Furthermore, we also explored the influence of the input power and liquid supply mode on the atomization performance and the heat generation ability of the atomization process to show the feasibility and performance advantages of the PSLEA atomization mode. Finally, based on the PSLEA atomization method, a handheld portable atomizer was designed for the first time. Its particle size showed a unimodal distribution, with a median particle size of 3.95 μm, which can meet the needs of lung inhalation therapy and lays a foundation for the commercial application of SAW atomization.

## 2. Experimental Setup and Method

### 2.1. SAW Devices

The Rayleigh SAW chip, with dimensions of 10 mm × 8 mm × 0.5 mm, is fabricated using a low-loss piezoelectric substrate: 0.5 mm thick 128° Y-cut, X-propagating single-crystal lithium niobate (LiNbO_3_) wafer. A 5 nm titanium/150 nm aluminum interdigital transducer (IDT) is patterned on the LiNbO_3_ substrate via standard photolithography and lift-off techniques, as depicted in Figure 1a. The straight-finger IDTs adopt a standard linear structure with equal electrode width and spacing (both 33.25 μm), 30 electrode pairs, and a 4 mm aperture. Consequently, the SAW wavelength (λ) o is 133 μm (four times the electrode width), and the excited SAW response frequency (*f_r_*) is 30 MHz, calculated by the relationship *f_r_* = *c*_s_/λ, where *c*_s_ ≈ 3990 m/s is the characteristic phase velocity of Rayleigh wave propagation along the 128° Y-X LiNbO_3_ substrate [21]. To extend the lifetime of the SAW atomizer, we fixed the SAW chip on the upper surface of an aluminum heat sink with dimensions of 30 mm × 12 mm × 4 mm using nano silver adhesive, as detailed in our previous literature [22]. Additionally, a 0.5 mm thick printed circuit board (PCB) with two pads is attached to the aluminum heat sink parallel to the SAW chip. The pads are electrically connected with the bus electrode of the IDT through conductive silver glue, and an RF signal matching the operating frequency of the SAW device is input into the IDT to excite SAWs on the surface of the piezoelectric substrate. The measured frequency response curve of the fabricated SAW atomizer by a network analyzer (E5063A, Keysight, Santa Rosa, CA, USA) is shown in Figure 1b, showing that the actual operating frequency of the device is 29.95 MHz and the return loss is about −20 dBm. Although slight deviations exist between the actual operating frequency and the theoretical design value due to inherent fabrication process variations, their impact on experimental research is negligible and can be ignored.

### 2.2. Experimental Setup

To achieve continuous liquid supply, a rectangular polyester–cellulose fiber strip (C1, LymTech, Chicopee, MA, USA) was employed as a capillary transport channel. One end of the paper strip was immersed in a deionized water container, while the other end was in contact with the SAW atomizer. As shown in Figure 1c, the front end of the liquid supply paper strip is located at the edge of the atomizer, which we call the PSLEA atomization mode. As the control group, the traditional atomization method, where the liquid supply paper strip is placed directly on the atomizer surface, as shown in Figure 1d, is designated as the PSLSA atomization mode. In the experiment, the PSLEA atomization configuration was systematically compared with the conventional PSLSA atomization configuration to evaluate its performance in the SAW atomization process.

In the experiments, an alternating current (AC) electrical signal matching the operating frequency of the SAW atomizer was generated by a radio frequency (RF) signal generator (DSG3030, RIGOL, Suzhou, China) and amplified by a power amplifier (Mini-circuit LZY-22). The SAW atomization process was dynamically monitored and recorded using a high-speed camera (CP70-1 HS-M-1900, Optronics, Kehl, Germany) with a frame rate of up to 40,000 frames per second (fps) and an ultramicroscope lens (LW-FF 25 mmf/2.8 ULTRA MACRO 2.5-5.0X, LAOWA, Hong Kong, China). An infrared thermal imaging instrument (A655SC, FLIR, Wilsonville, OR, USA) with a resolution of 640 × 480 pixels and a frame rate of up to 200 Hz, equipped with a 25 μm macro lens, was employed to record the temperature field distribution in the atomization zone instantaneously. The aerosol generation rate was accurately estimated by measuring the total mass loss per unit time using a precision electronic balance (FA2004B, JK, Shanghai, China) connected to a computer, providing an average atomization rate over a specified period based on the slope of the mass–time graph to support the quantitative analysis. A laser diffraction analyzer (DP-02, OMCC, Zhuhai, China) was used to measure the size distribution of the atomized aerosols at a height of 20 mm above the atomizer surface. Each group was tested continuously three times, with each test lasting for 15 s, and the average value of the three tests was taken as the result for that group. In addition, the thermal stress distribution in the atomization zone was calculated using multi-physical field simulation software (COMSOL Multiphysics 6.2, Stockholm, Sweden) based on the temperature data extracted from the infrared thermal imager.

## 3. Results and Discussion

### 3.1. Acoustic Wetting and Atomization Process

The key frames of the deionized water atomization process in two atomization modes under an input voltage of 800 mV are recorded by high-speed image equipment, as shown in Figure 2a and Figure 2b, respectively. The green hollow arrow indicates the droplet jetting direction, and the red hollow arrow indicates the aerosol atomization direction. Firstly, the liquid in the paper strip is pulled out and stretched in the direction opposite to SAW propagation to form a thin liquid film, a phenomenon called the acoustic wetting effect. In the PSLEA atomization mode, acoustic wetting only generates an extremely narrow liquid film at the edge where the paper strip contacts the edge of the atomizer. Although the droplet jetting phenomenon is observed at about 0.5 ms, the front end of the paper strip remains intact without any deformation. Then, the PSLEA atomization mode simultaneously produces droplet jetting and aerosol atomization at about 1.5 ms. The droplet jetting direction is essentially consistent with the Rayleigh angle, while the aerosol atomization direction is nearly perpendicular to the atomizer surface, forming a clear separation between the atomization and jetting direction. Over time, the concentration of atomized aerosol and the number of jetting droplets increase synchronously, while the separation of atomization and jetting direction is consistently maintained. As the narrow liquid film at the edge of the atomizer is gradually consumed, the number of spray droplets decreases. At approximately 460 ms, the droplet jetting phenomenon nearly disappears completely, and the atomization process stabilizes, generating only submicron-sized aerosol plumes that move vertically upward at high velocity.

When the PSLSA atomization mode is adopted, the front end of the paper strip is slightly lifted by the acoustic wetting effect, causing the liquid to accumulate at the front end and form a thicker liquid film than the paper strip itself. Then, the liquid accumulated at the front of the paper strip gradually forms an inclined liquid column jetting at about 58 ms. As the inclined liquid column is stretched by diffractive acoustic waves, it is gradually pinched off into multiple liquid droplets along the Rayleigh angle at about 100 ms. As the liquid is consumed, the liquid film at the front end of the paper strip gradually expands and thins toward the IDTs, and a mixed state of atomized aerosol and jetting droplets with nearly coincident inclination angles emerges at about 380 ms. Throughout the atomization process, the atomized aerosol plume exhibits a relatively dispersed distribution and remains accompanied by droplet jetting. Although the number of jetting droplets decreases as atomization proceeds, separating the atomized aerosol from the jetting droplets is tricky.

### 3.2. The Effect of the Input Power

Based on the atomization process of deionized water under the two atomization modes, the different atomization performances mainly originate from the morphology and dimensions of the liquid film at the front end of the paper strip. These characteristics directly influence the atomization efficiency, particle size distribution, atomization form, and stability. The following analysis investigates the atomization performance under different input power conditions, in which the input voltage range is set between 600 mV and 800 mV to ensure typical liquid atomization in both modes effectively. As shown in Figure 3a, the upper and lower rows of the images depict the morphological changes in the liquid film at the front end of the paper strip when the PSLEA and PSLSA atomization reaches stable atomization under different input voltages, respectively. The red rectangular boxes highlight the liquid film area. Here, the dimension of the liquid film along the acoustic wave propagation direction is defined as the liquid film length, and the dimension along the IDT electrode is defined as the liquid film width. The results show that the liquid film width is essentially consistent with the IDT aperture, while the liquid film length distribution differs significantly between modes. As shown in Figure 3c, the liquid film length distribution under different input voltages shows that the liquid film length gradually decreases with increasing input power in both atomization modes. When the input voltage increases from 600 mV to 800 mV, the liquid film length in the PSLSA mode decreases from about 3 mm to 1.2 mm, whereas that in the PSLEA mode decreases from about 1 mm to 0.4 mm. Notably, the liquid film length in the PSLEA mode is reduced by half compared to the PSLSA mode under the same input power. On the one hand, the SAW energy also increases with input power. The enhanced acoustic force generates a volume flux in the fluid along the direction of acoustic radiation, which acts on the liquid membrane and pushes it toward the paper strip, thereby promoting film contraction. On the other hand, the expansion of the liquid film under acoustic wetting is partially constrained by gravity. When the liquid is located at the edge of the atomizer, the expansion speed of the liquid film is further restricted by gravitational pull and acoustic wave attenuation at the edge, which more effectively drives liquid film length contraction.

In the PSLEA atomization mode, a shorter liquid film shortens the acoustic wave–liquid coupling path, effectively reducing acoustic wave energy attenuation during propagation in the liquid film. This allows the acoustic wave energy to act on the liquid surface more intensively and efficiently, making the atomization aerosol on the liquid film surface more directional and concentrated. As shown in Figure 3b, close-up shots of aerosols in two atomization modes under an input voltage of 800 mV were recorded. The results show that a stable aerosol plume with dimensions of approximately 1.8 mm in the X direction and 4 mm in the Y direction, essentially perpendicular to the atomizer surface, can be achieved at a height of 10 mm above the atomizer surface for the PSLEA mode. In contrast, the PSLSA atomization mode produced a mixed aerosol–droplet plume with dimensions of approximately 8 mm in the X direction and 5.2 mm in the Y direction at the same height along a significant inclination angle relative to the atomizer surface. An excessively long liquid film may induce irregular fluctuations on the liquid surface, leading to inconsistent droplet sizes and even the formation of irregular large droplet jetting. The shorter liquid film in the PSLEA mode concentrates acoustic energy and promotes regular liquid surface fluctuations, facilitating the formation of more uniform aerosols. Compared to the PSLSA mode, the PSLEA mode generates a more directional, regular, uniform, and convergent aerosol plume, offering advantages in improving spray quality, optimizing drug delivery efficiency, and reducing material waste.

In addition, the shorter liquid film length in the PSLEA mode enables rapid adaptation to changes in SAW excitation, thereby facilitating rapid startup and shutdown of the atomization process to enhance the response speed of the system significantly. As shown in Figure 3d, when the input voltage increases from 600 mV to 800 mV, the time required to achieve stable atomization decreases from approximately 46 s to about 1 s, demonstrating a nearly exponential decay trend. This implies that by selecting an appropriate power level, a stable, droplet-free, and directionally convergent aerosol atomization state can be achieved instantaneously at startup, enabling immediate atomization. Although Figure 3b shows that the PSLSA mode produces a broader and more dispersed aerosol distribution at the same height above the atomizer surface under identical power conditions, actual testing of the liquid mass changes before and after atomization under different power conditions, and the calculation of the corresponding atomization rate yielded surprising results. As shown in Figure 3e, the atomization rates of both atomization modes increase significantly with increasing input power. However, the atomization rate is significantly higher for PSLEA under the same input power conditions. When the input voltage increases from 600 mV to 800 mV, the atomization rate of deionized water in the PSLSA mode increases from about 0.2 mL/min to 1.3 mL/min, while the PSLEA mode shows an increase from about 1.1 mL/min to 1.62 mL/min. At a low-power threshold, the advantage of the PSLEA mode is striking, with a nearly 5.5-fold difference in atomization rate compared to the PSLAS mode. Although the difference narrows at a high-power threshold, it also remains at approximately 1.23-fold. The reason can be attributed to the following mechanism: In the PSLSA mode, the longer liquid film and larger acoustic wave–liquid contact area extend the atomization zone, reducing the density of acoustic wave energy applied to the liquid and resulting in lower initial aerosol velocities. At high power, the PSLSA mode exhibits more intense droplet jetting phenomena, producing a greater quantity of large-sized droplets. This leads to a pronounced increase in the atomization rate, thereby narrowing the performance gap with the PSLEA mode. However, the phenomenon of more large-sized droplet jetting can further reduce the effective utilization rate of aerosols.

### 3.3. The Effect of the Liquid Supply Mode

In the SAW atomization process, the liquid supply rate, though often overlooked, is one of the important parameters that affect the atomization performance of the liquid. It is closely related to the quality of liquid atomization, the droplet size distribution characteristics, and aerosol generation efficiency. Due to the unique coupling effect between fluid flow and acoustic streaming during the atomization process, adjusting the liquid supply rate can exert complex and significant impacts on atomization quality. As shown in Figure 4a, we employed five different paper strip combination liquid supply methods, designated as #1, #2, #3, #4, and #5. All paper strips have a width of 12 mm, matching the width of the atomization chip. Specifically, #1, #2, and #3 represent single-layer paper strips, double-layer stepped distribution paper strips, and triple-layer stepped distribution paper strips, respectively. These form the first control group. In the PSLEA atomization mode, the liquid supply height at the front end of the paper strip is fundamentally consistent with the thickness of the atomizer chip, while the stepped distribution design introduces variations in the liquid supply rate. Synchronously, #1, #4, and #5 constitute the second group, where #4 and #5 are double-layer and triple-layer laminated paper strips, respectively. The laminated structure elevates the liquid supply height at the front end of the paper strip above the thickness of the atomizer chip, simultaneously introducing two variables: liquid supply rate and liquid supply height difference.

For both control groups, PSLEA and PSLSA atomization experiments with deionized water were conducted at an input voltage of 800 mV. The morphologies of the liquid film in the corresponding atomization zones are shown in Figure 4b,c, and the corresponding liquid film length distribution is shown in Figure 4d. It can be observed that the liquid film size varies significantly with changes in the liquid supply mode of the paper strip combination. In the first control group, the liquid film length exhibits minor changes. Specifically, the liquid film length increases from about 0.4 mm to 0.8 mm in the PSLEA atomization mode and from about 0.8 mm to 2.3 mm in the PSLSA atomization mode. However, the liquid film length changes significantly in the second control group. Under double-layer and triple-layer laminated paper strips, the liquid film lengths in the PSLEA atomization mode are 1.5 mm and 2.6 mm, while those in the PSLSA atomization mode are 3 mm and 4.3 mm, respectively. The experimental results show that the liquid film size in the PSLEA atomization mode has been significantly expanded under the action of laminated paper strips, mimicking the characteristics of the PSLSA atomization mode. This expansion is closely correlated with the number of paper strip layers and their height. When the height of the paper strip exceeds the thickness of the atomizer chip, liquid accumulates on the atomizer surface, diminishing the unique advantages of the PSLEA atomization mode. Figure 4e depicts the atomization rate distribution of deionized water under different paper strip combinations for liquid supply. Although the liquid film size changes very little in the first control group, the liquid supply rate is effectively supplemented by the stepped distribution paper strips, further improving the atomization rate. For the PSLEA atomization mode, the atomization rates increase to about 2.3 mL/min and 2.6 mL/min under double-stepped and triple-stepped distribution paper strips, respectively. Due to the superior atomization capability of the PSLEA mode, the insufficient liquid supply rate of single-layer paper strips restricts the atomization rate. To address this limitation, the stepped configuration enhances the liquid supply rate by expanding the liquid supply area while maintaining a constant front-end height, thereby boosting the liquid atomization rate. However, in the second control group, the increase in liquid supply height causes severe liquid accumulation at the laminated paper strip front end, thereby increasing the resistance of the liquid film being pushed back toward the paper strip by the acoustic volumetric force. The atomization behavior in the PSLEA mode becomes similar to that of the PSLSA mode, where the atomized aerosol plume no longer maintains a vertical direction with large-sized droplets generated, compromising atomization quality and aerosol generation efficiency. Thus, when using PSLEA atomization, it is critical to ensure that the liquid supply height of the paper strip is not higher than the thickness of the atomization chip to form an optimal liquid film size. On this basis, combining a stepped distribution of paper strips to improve the liquid supply rate can further enhance the advantages of PSLEA atomization.

In addition to the liquid supply height at the front of the paper strip, the liquid supply width also significantly influences the liquid atomization performance. As shown in Figure 5a, we use four different widths of single-layer paper strips to supply liquid for PSLEA atomization at an input voltage of 800 mV, with the paper strip widths set at 2 mm, 4 mm, 8 mm, and 12 mm. This arrangement is deliberate, as the IDT aperture in the SAW atomizer is 4 mm. Under the four paper strip widths, both the length and width of the liquid film at the front end of the paper strip are extended with increasing paper strip widths, as shown in Figure 5b. Overall, the variation in the liquid film length is minimal (0.4 mm–0.5 mm), while the variation in liquid film width is considerably more significant (2 mm–4 mm). The atomization rate distribution of deionized water, shown in Figure 5c, demonstrates a positive correlation between paper strip width and atomization rate. The atomization rate increases significantly when the paper strip width increases from 2 mm to 4 mm. However, the growth trend gradually diminishes as the width increases from 4 mm to 8 mm. When the paper strip width is narrower than the IDT aperture (≤4 mm), the liquid supply capacity is limited, failing to maintain uniform and continuous replenishment to the acoustic beam area. The contact width between the liquid and the SAW is narrow, which limits the energy utilization efficiency of the SAW and the corresponding atomization rate. When the width of the liquid supply paper exceeds the IDT aperture (>4 mm), the paper strip width fully covers the aperture range. At this point, the paper strip partially located outside the aperture can effectively supplement the liquid supply, ensuring that the interaction width between the SAW and the liquid film is equivalent to the IDT aperture size so that the acoustic wave energy can be fully transmitted to the liquid film surface, effectively improving the liquid atomization rate.

### 3.4. Heat Generation Capacity Analysis

In the SAW atomization process, attention should be paid not only to aerosol quality, droplet size distribution characteristics, and atomization efficiency but also to the heat generation capacity of the liquid under SAW excitation. Excessive temperature rise can rapidly degrade the atomization chip and compromise functional liquids, limiting biomedicine and pharmaceutical atomization applications. The atomization process of the deionized water under two atomization modes was observed in real time using an infrared camera to record the dynamic change and equilibrium state of the liquid temperature for a single-layer paper strip with a width of 12 mm and an input voltage of 800 mV. The temperature distributions at the stable equilibrium state under the PSLEA and PSLSA atomization modes are shown in Figure 6a and Figure 6b, respectively. The white rectangular boxes in the figure identify the atomization zone and display the corresponding temperature values, where the first datum represents the maximum temperature value and the second datum represents the average temperature value. The results show that the PALEA atomization mode exhibits about a 20 °C reduction in maximum temperature and about a 15 °C reduction in average temperature compared to the PSLSA atomization mode. On the one hand, in the PSLEA atomization mode, the shorter liquid film at the edge of the atomization chip minimizes the liquid–atomizer contact areas, shortening the acoustic energy dissipation path within the liquid. Consequently, more acoustic energy is concentrated on liquid atomization rather than heat generation, thus reducing local heat accumulation in the liquid film. On the other hand, the higher atomization rate in the PSLEA atomization mode enables the atomized aerosol to rapidly carry away part of the heat, further reducing the heat accumulation at the liquid film interface.

A scatter diagram of the maximum temperature at each moment during the deionized water atomization process and the corresponding temperature fitting curve are shown in Figure 6c, which indicates that the temperature rise in the SAW atomization process can be divided into two distinct stages. In the first stage, the temperature in the atomization zone rises rapidly within 1 s of the atomizer startup, primarily attributed to the viscous dissipation of SAW energy in the liquid. In the second stage, the temperature rise rate in the atomization region gradually slows and stabilizes. In this stage, heat conduction between the liquid and the atomization chip and heat convection between the liquid/solid and the air become dominant, making the heat exchange more sufficient for finally reaching dynamic equilibrium. Since a specific temperature difference exists between the atomization zone and other areas on the atomization chip surface, thermal stress will be generated in the chip material. The accumulated thermal stress may cause material fatigue and deformation, potentially leading to damage or failure of the atomizer equipment.

To validate the above speculation, the finite element analysis (FEA) method was employed to establish a surface heat transfer and thermal stress analysis model for the atomizer, as shown in Figure 6d. This model mainly includes a piezoelectric substrate (128° YX LiNbO_3_) and the liquid film in the atomization zone on the upper right surface of the substrate. The size of the piezoelectric substrate is 10 mm × 4 mm × 0.5 mm, and the material properties refer to the data in the literature [23]. The liquid film dimension is 0.5 mm × 4 mm × 0.01 mm for the PSLEA atomization mode and 4 mm × 4 mm × 0.01 mm for the PSLSA atomization mode. During the simulation process, the fitting curve of the measured temperature in the atomization zone was used as the temperature input function of the heat source for the surface liquid film. The bottom of the model was set as heat conduction, with a thermal conductivity of 10 W/(m^2^·k), and all other surfaces were set as natural convection boundaries. Then, a transient solver was used to calculate the temperature and thermal stress distribution on the atomizer surface. Figure 6e,f show the temperature distributions for PSLEA and PSLSA atomization in the equilibrium state through simulation calculations, indicating that the highest temperature point is consistently located at the front end of the liquid film. Figure 6g shows that the simulation-predicted maximum temperatures in the atomization zone align with the experimental results over a 20-s interval. Despite minor fluctuations in the experimental results for external environmental factors, the overall trend is entirely consistent, confirming the accuracy of the FEA model.

The vertical component distribution curve of the thermal stress tensor within the acoustic transmission path on the atomizer surface is shown in Figure 6h. The results indicate sharp thermal stress tensor fluctuations near the boundary between the liquid film and the substrate, where stress-induced cracking and failure of the atomization chip are most likely to occur. The temporal evolution of the maximum stress value is shown in Figure 6i, in which it increases initially and then gradually decreases with time. The maximum stress value occurs at about 0.3 s for the PSLEA atomization mode and 0.7 s for the PSLSA atomization mode, corresponding to the turning point of the temperature rise from the first stage to the second stage in Figure 6c. After that, accelerated heat exchange slows the heating rate, improves the temperature uniformity on the surface of the atomizer, and causes the thermal stress to decrease gradually. Due to the difference in the position and dimension of the liquid film on the surface of the atomizer under the two atomization modes, the maximum thermal stress point for the PSLEA atomization mode is close to the atomizer edge with a value of about 4.3 × 10^8^ N/m^2^, while the maximum thermal stress point for the PSLSA atomization mode is far away from the chip edge, with a value of about 7.9 × 10^8^ N/m^2^. On the one hand, the maximum thermal stress in the PSLEA atomization mode is about 45% lower than in the PSLSA atomization mode. On the other hand, stress near the chip edge is more easily released, minimizing the risk of atomizer damage. Through experimental tests, when deionized water is atomized by the PSLEA atomization method under the input voltage condition of 800 mV, no phenomenon of atomizer chip cracking occurs, effectively protecting the atomization chip. Table 1 summarizes the atomization characteristics between PSLEA and PSLSA with deionized water supply from a single-layer paper strip and an input voltage of 800 mV. Under the same conditions, the atomization performances that can be achieved by the PSLEA mode are superior to those of the PSLSA mode, including a faster atomization rate, a narrower vertical aerosol plume size, a lower maximum temperature, and less thermal stress.

### 3.5. Handheld Atomizer Based on Edge Atomization

With the continuous development of medical technology, aerosol inhalation administration has emerged as a critical and practical mainstream approach to treating respiratory diseases. Traditional medical atomizers, such as compression atomizers and mesh atomizers, still exhibit notable limitations. Compression atomizers suffer from high noise, bulkiness, poor portability, and reliance on external power sources, restricting their usage scenarios. Mesh atomizers face challenges like easy clogging of mesh holes, low compatibility with medicinal solutions, and high price, limiting their popularity. In recent years, many researchers have attempted to explore the application of SAW atomization technology in handheld medical atomizers. However, persistent issues, such as the uneven particle size distribution of atomized aerosol and unstable atomization direction, have hindered previous efforts, which pose significant barriers to practical medical applications. In this paper, by introducing the PSLEA atomization method, the above problems can be effectively solved, and a handheld and portable atomizer with good spray quality, uniform droplet size distribution, and high aerosol generation efficiency can be obtained.

A handheld portable SAW atomizer based on the PSLEA atomization method is designed and presented in Figure 7a. This SAW atomizer mainly consists of the atomization cup and the power supply. The atomization cup is mainly composed of an SAW atomization chip, a porous liquid conducting medium, a liquid storage chamber, and sealing elements, as depicted in Figure 7c, and the power supply is mainly composed of a driving circuit board (Figure 7d) and a battery (Figure 7e). The liquid to be atomized is filled into the liquid storage chamber, and then, the porous liquid guiding medium is used to guide the liquid to the edge of the SAW atomization chip. Upon activating the power supply, the atomization function of the liquid can be achieved. This atomizer does not rely on the movement of mechanical components or the propulsion of airflow, making it suitable for long-term use by the elderly and infants without mechanical friction or air flow noise. The particle size distribution of atomized deionized water aerosol was measured by a laser particle size analyzer, as shown in Figure 7b, where the particle size presents a unimodal distribution, with a median particle size of 3.95 μm, meeting the requirements for pulmonary inhalation treatment. During sustained operation tests, this SAW atomizer produced a stable mist output with no noticeable performance degradation as the liquid volume in the atomization cup decreased, demonstrating its capability for efficiently delivering medicinal solutions to the deep lung regions.

## 4. Conclusions

In this study, we present a sustainable and efficient atomization method with a liquid supply paper strip located at the edge of an SAW atomizer. Under the effect of acoustic wetting, this method forms a micro-scale narrow liquid film at the junction between the atomizer edge and the paper strip. At the initial startup of the SAW atomizer, it simultaneously generates atomized aerosol perpendicular to the atomizer surface and droplet jetting at the Rayleigh angle, effectively achieving physical separation of the aerosol and large droplets. As the thin liquid film stabilizes relatively quickly, the droplet jetting phenomenon almost wholly disappears, ultimately forming an ideal aerosol plume characterized by uniform particle size distribution, directional concentration, and stable convergence. The effect of input power on the atomization performance, including liquid film length, characteristic size of the aerosol plume, stable atomization time, and atomization rate, was investigated in detail through experiments. The results showed that as the input power increases, the liquid film length decreases linearly, the stable atomization time decays exponentially, and the atomization rate gradually increases. Specifically, under the same input power condition, the PSLEA atomization mode exhibits approximately 50% shorter liquid film lengths and up to 5.5-fold higher atomization rates compared to the PSLSA atomization mode. The influence of factors such as the liquid supply rate, the height difference of the liquid supply, and the width of the liquid supply on the atomization performance was further explored through different liquid supply methods with combinations of paper strips. The results show that the change in the dimension of the liquid film is tiny with the stepped paper strip for liquid supply, but the liquid supply rate is effectively supplemented, which can further increase the atomization rate. Under an input voltage of 800 mV, the atomization rate increased from 1.6 mL/min (single-layer paper strip) to 2.3 mL/min (two-layer stepped distribution paper strip) and 2.6 mL/min (three-layer stepped distribution paper strip). When the liquid supply height exceeds the atomizer chip thickness using the laminated paper strip in the PSLEA mode, severe liquid accumulation at the paper strip front end enlarges the liquid film significantly, substantially decreasing the atomization rate and making its characteristics approach those of PSLSA. In addition, to ensure the liquid supply capacity and the utilization rate of acoustic energy, the width of the liquid supply by the paper strip needs to be equal to or slightly larger than the IDT aperture of the atomization chip. Then, an infrared thermal imager was used to record the dynamic changes and equilibrium state of the liquid temperature in the atomization zone under the action of an SAW. The results showed that the maximum temperature in the PSLEA atomization mode is reduced by about 20 °C, and the average temperature is reduced by about 15 °C compared with the traditional atomization mode. Correspondingly, the calculated maximum thermal stress value in the PSLEA atomization mode is approximately 45% lower than in the PSLSA atomization mode, dramatically reducing the failure rate of the atomization chip. Finally, based on the PSLEA atomization method, we successfully developed a portable handheld atomizer with stable performance. The atomized aerosol exhibits a unimodal particle size distribution with a median diameter of 3.95 μm, meeting the requirements for pulmonary inhalation therapy and making the commercial application of SAW atomizers in treating respiratory diseases possible.

## Figures and Tables

**Figure 1 micromachines-16-00628-f001:**
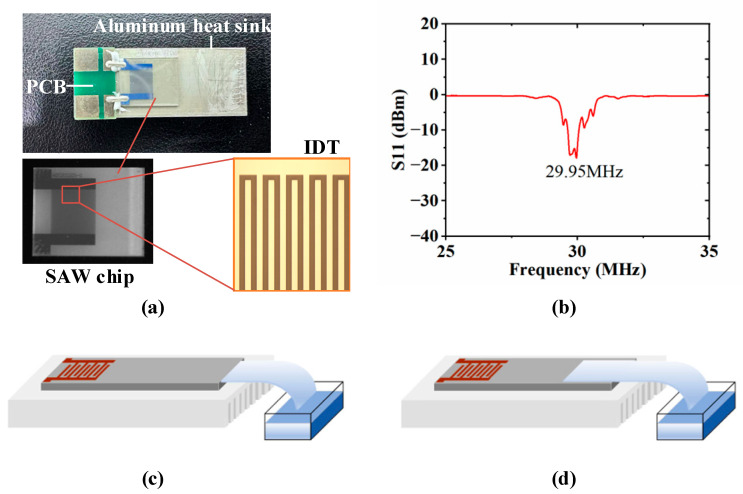
Atomization experiment equipment and setup. (**a**) SAW devices, (**b**) the frequency response curve of the fabricated SAW devices, (**c**) schematic of PSLEA atomization, (**d**) schematic diagram of conventional PSLSA atomization.

**Figure 2 micromachines-16-00628-f002:**
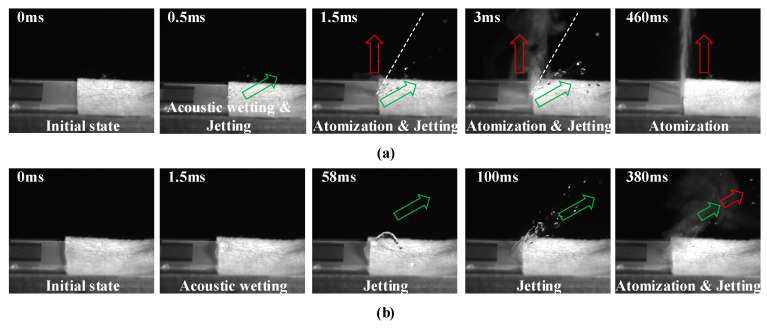
Atomization process of deionized water supplied by a paper strip with an input voltage of 800 mV. (**a**) PSLEA atomization mode, (**b**) PSLSA atomization mode. The green hollow arrow indicates the droplet jetting direction, and the red hollow arrow indicates the atomization direction.

**Figure 3 micromachines-16-00628-f003:**
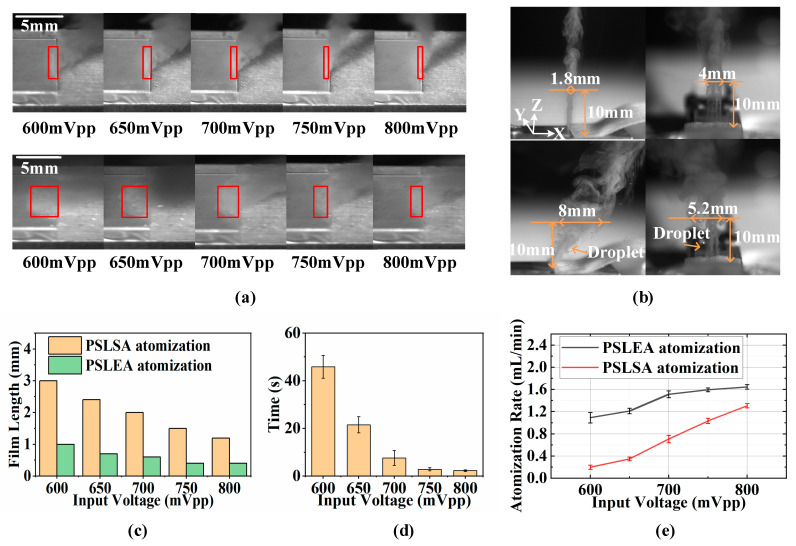
The influence law of input power on atomization characteristics. (**a**) The morphological changes in the liquid film at the front end of the paper strip under different input voltages, (**b**) overview of the atomized aerosol under an input voltage of 800 mV, (**c**) the distribution of liquid film length under different input voltages, (**d**) the relationship between the time required for stable atomization and input voltage in the PSLEA atomization mode, (**e**) the relationship between the atomization rate of deionized water and the input voltage. The red rectangular boxes indicate the liquid film area at the front end of the paper strip.

**Figure 4 micromachines-16-00628-f004:**
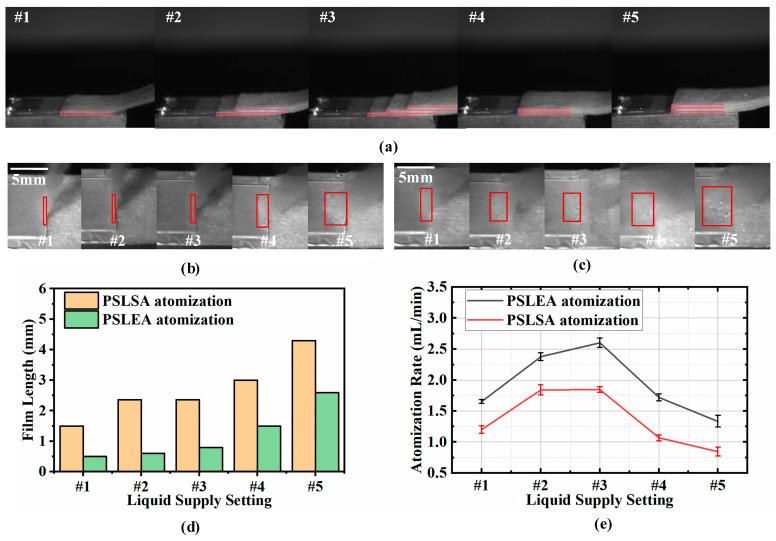
The influence of five different combinations of paper strips for liquid supply on atomization performance. (**a**) A schematic diagram of the paper strip combination for liquid supply, (**b**) a feature image of the liquid film at the atomization zone under PSLEA atomization, (**c**) a feature image of the liquid film at the atomization zone under PSLSA atomization, (**d**) the distribution of liquid film lengths, (**e**) the distribution of atomization rates. The red lines describe the thickness contour of the liquid supply paper strip positioned on the atomizer surface. The red rectangular boxes indicate the liquid film area at the front end of the paper strip.

**Figure 5 micromachines-16-00628-f005:**
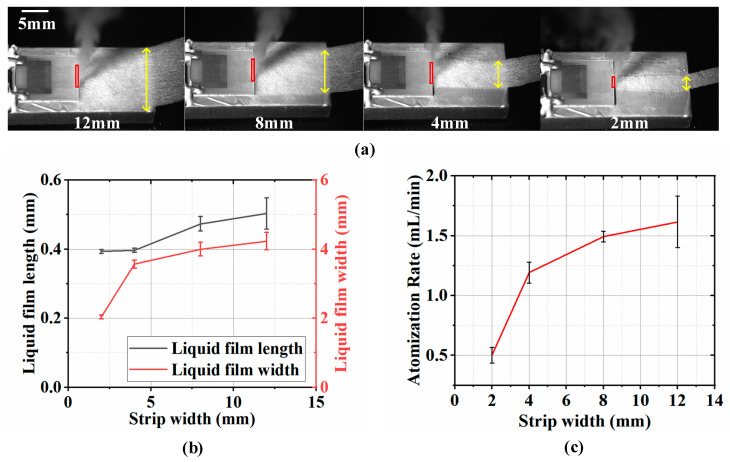
The influence of the liquid supply width of the paper strips on atomization performance. (**a**) A schematic diagram of the liquid supply for PSLEA atomization using a single-layer paper strip with four different widths, (**b**) the liquid film length and width varies with the paper strip width, and (**c**) the atomization rate varies with the paper strip width. The red rectangular boxes indicate the liquid film area at the front end of the paper strip. The yellow lines with double arrows denote the width of the liquid supply paper strip.

**Figure 6 micromachines-16-00628-f006:**
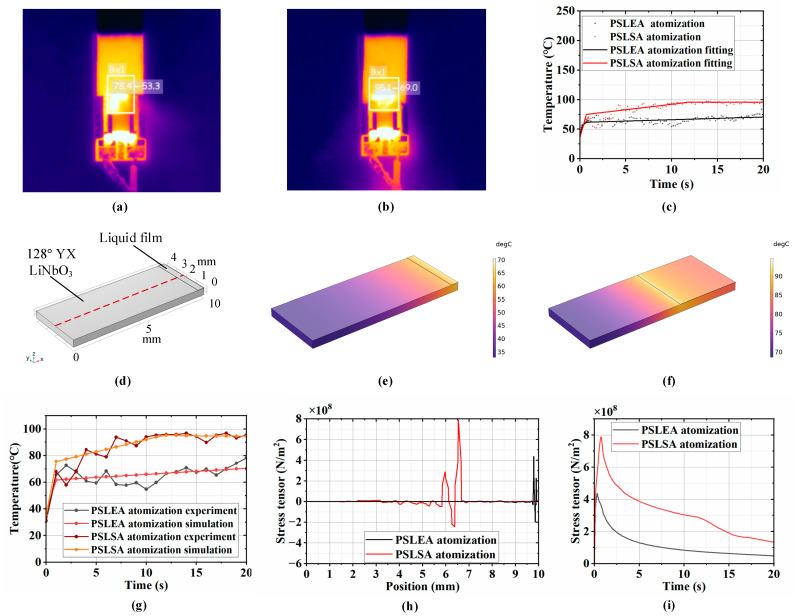
The heat generation capacity at the atomization zone. (**a**) Temperature distribution diagram at a stable equilibrium state under PSLEA atomization, (**b**) temperature distribution diagram at a stable equilibrium state under PSLSA atomization, (**c**) the maximum temperature scatter diagram and corresponding temperature fitting curve of deionized water at each time in the atomization process, (**d**) surface heat transfer and thermal stress analysis model, (**e**,**f**) temperature distribution of PSLEA atomization and PSLSA atomization under the simulation model, (**g**) corresponding simulation results and comparison with the experimental results, (**h**) stress tensor distribution of the central line on the substrate surface, (**i**) maximum stress tensor varies with time.

**Figure 7 micromachines-16-00628-f007:**
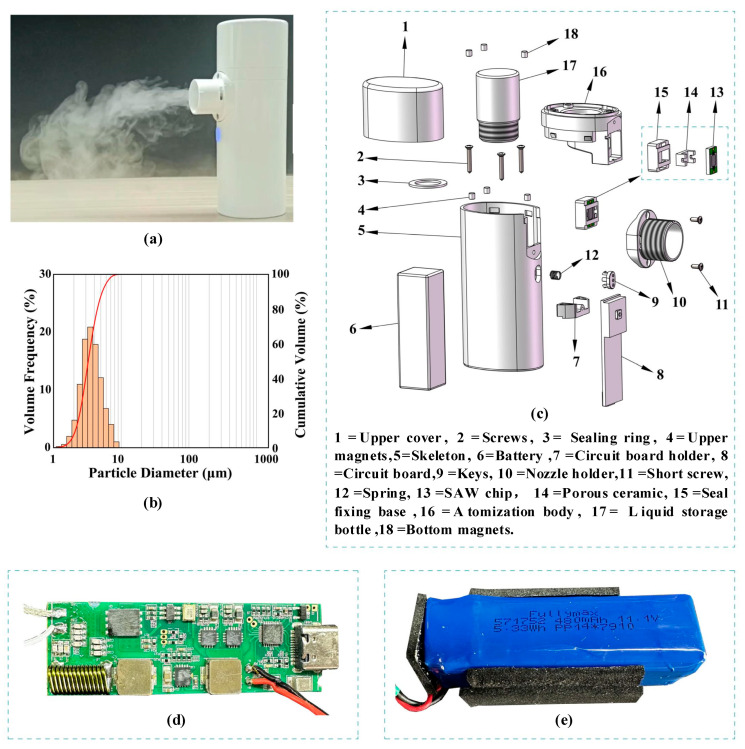
Handheld portable SAW atomizer based on the PSLEA atomization method. (**a**) A physical map of the SAW atomizer, (**b**) the particle size distribution of atomized aerosol in deionized water, (**c**) a structure-exploded view of the SAW atomizer, (**d**) the driving circuit board, and (**e**) a battery.

**Table 1 micromachines-16-00628-t001:** Comparison of atomization performances between PSLEA and PSLSA with deionized water supply from a single-layer paper strip and an input voltage of 800 mV.

	PSLEA	PSLSA
Atomization speed	1.62 mL/min	1.30 mL/min
Aerosol angle	90°	67°
Aerosol plume size	1.8 mm × 4 mm	8 mm × 5.2 mm
Liquid film length	0.4 mm	1.2 mm
Maximum temperature	78.4 °C	95.1 °C
Maximum thermal stress	4.3 × 10^8^ N/m^2^	7.9 × 10^8^ N/m^2^

## Data Availability

The data will be made available on request.

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
