# Peer review of "Low Thermal Stress and Instant Efficient Atomization of Narrow Viscous Microfluid Film Using a Paper Strip Located at the Edge of a Surface Acoustic Wave Atomizer"

_micromachines, 2025, doi:10.3390/mi16060628_

Round 1
Reviewer 1 Report
Comments and Suggestions for Authors
The authors proposed an atomization method in which a paper strip is placed at the edge of the atomizer (PSLEA). This configuration forms a micron-sized narrow liquid film at the junction between the edge of the atomization chip and the paper strip under the influence of acoustic wetting.
In the handheld portable system, what are the details of the driving circuit design? How did the authors address the issue of white noise in the circuit?
Although the experimental data was well presented, additional details regarding the portable system should be included. This would significantly enhance the paper’s impact in the field.
It would be beneficial to include a table summarizing the evaluation results for each performance metric—not only comparing PSLEA and PSLSA, but also including other potential approaches.
Overall, the authors clearly demonstrated the proposed concept, including the implementation of the portable system.
Comments on the Quality of English LanguageNA
Author Response
Comment1:In the handheld portable system, what are the details of the driving circuit design? How did the authors address the issue of white noise in the circuit?
Response1:We acknowledge your inquiry about the driving circuit design. In addition to SAW devices and liquid supply, the driving circuit is indeed a key focus in achieving handheld portable system. The driving circuit is only used to replace the signal source and power amplifier used in experiments, providing frequency matched sinusoidal electrical signals for SAW chips in handheld portable system. As this study focuses primarily on the atomization mechanism and microfluidic film dynamics enabled by the PSLEA configuration, the circuit design was intentionally simplified. The detailed circuit design is in the field of electronic technology, and details can be found in our relevant patent documents (CN202110204452.1, CN202411540444.4, CN202411535443.0).
Comment2:Although the experimental data was well presented, additional details regarding the portable system should be included. This would significantly enhance the paper’s impact in the field.
Response2:Thanks for your sincere suggestion. This article mainly focuses on the “atomization characteristics of narrow viscous microfluid film with paper strip located at the edge of the SAW atomizer”. The realized portable system in the last section of this paper is used to demonstrate the feasibility of the PSLEA method proposed in this article. Details regarding the portable system can be obtained from our publicly available patents (CN202410612965.X).
Comment3:It would be beneficial to include a table summarizing the evaluation results for each performance metric—not only comparing PSLEA and PSLSA, but also including other potential approaches.
Response3:Thanks for your sincere suggestions. At the end of Section 3.4 of the article, we added Table 1, which summarizes a detailed comparison of the atomization performances between PSLEA and PSLSA. Unfortunately, the efficiency of liquid supply through capillary tube and capillary slit mentioned in the Introduction is low and lack of detailed data support, so those two potential approaches were not included in the table. The widely recognized and used method for SAW atomization currently is liquid supply through a paper strip to achieve continuous liquid atomization. A detailed comparison of the atomization performance of the two modes under the same conditions can highlight the advantages of the PSLEA mode proposed in the article.
Special thanks to you for your valuable comments and suggestions.
Reviewer 2 Report
Comments and Suggestions for Authors
Dear Authors,
The proposed manuscript "Low thermal stress and instantly efficient atomization of nar-row viscous microfluid film with paper strip located at the edge of the SAW atomizer" has exibited great results, which can be accepted after minor adjusts. (i) Please, the authors need to adjust the utilized unit of measurement e.g. on page 17 "the atomization rate of 1.6ml/min". (ii) Some Figs has exibited low quality in terms of resolution e.g. Figs 7b, c and d. Also, Fig 7e has exhibited a mistake in the informed unit of measurement for particle diameter.
Author Response
Comment 1:Please, the authors need to adjust the utilized unit of measurement e.g. on page 17 "the atomization rate of 1.6ml/min".
Response 1:Thank you for pointing this out. We have revised all the units of "ml/min" to "mL/min" in the paper. Additionally, we have standardized all units of this paper according to the journal guidelines.
Comment 2:Some Figs has exibited low quality in terms of resolution e.g. Figs 7b, c and d. Also, Fig 7e has exhibited a mistake in the informed unit of measurement for particle diameter.
Response 2:Thank you for the issue you pointed out. All low-quality Figures in this paper have been re-imported to meet the publication requirements. Figures 7b, c and d have been redrawn and reformatted using software to obtain clearer images. The unit for particle diameter in Figure 7b has been corrected from "um" to "μm" in both the figure caption and the axis label.